**Data Availability Statement:** Secondary analysis was conducted of anonymised data collected by

# The prevalence and socio-demographic associations of household food insecurity in seven slum sites across Nigeria, Kenya, Pakistan, and Bangladesh. A cross-sectional study

**Clara Spieker**[1], **Anthony A. Laverty**[1]*, **Oyinlola Oyebode**[2], **The Improving Health in Slums Collaborative**[¶]

1 Imperial College London, London, United Kingdom, 2 Wolfson Institute of Population Health, Queen Mary University of London, London, United Kingdom

¶ Membership of The Improving Health in Slums Collaborative is provided in the Acknowledgments
* a.laverty@ic.ac.uk

## Abstract

Although the proportion of people living in slums is increasing in low- and middle-income countries and food insecurity is considered a severe hazard for health, there is little research on this topic. This study investigated and compared the prevalence and socio-demographic associations of household food insecurity in seven slum settings across Nigeria, Kenya, Pakistan, and Bangladesh. Data were taken from a cross-sectional, household-based, spatially referenced survey conducted between December 2018 and June 2020. Household characteristics and the extent and distribution of food insecurity across sites was established using descriptive statistics. Multivariable logistic regression of data in a pooled model including all slums (adjusting for slum site) and site-specific analyses were conducted. In total, a sample of 6,111 households were included. Forty-one per cent (2,671) of all households reported food insecurity, with varying levels between the different slums (9–69%). Household head working status and national wealth quintiles were consistently found to be associated with household food security in the pooled analysis (OR: 0·82; CI: 0·69–0·98 & OR: 0·65; CI: 0·57–0·75) and in the individual sites. Households which owned agricultural land (OR: 0·80; CI: 0·69–0·94) were less likely to report food insecurity. The association of the household head's migration status with food insecurity varied considerably between sites. We found a high prevalence of household food insecurity which varied across slum sites and household characteristics. Food security in slum settings needs context-specific interventions and further causal clarification.

## Introduction

Food insecurity is described by the Food and Agriculture Organization of the United Nations (FAO) as the 'lack [of] regular access to enough safe and nutritious food for normal growth

the NIHR Global Health Research Unit on Improving Health in Slums (https://warwick.ac.uk/fac/sci/med/about/centres/wcfgh/slums). Ethical approval allowed for anonymised data to be shared with researchers outside the specified Global Health Research Unit with the permission of the Unit's management and financial committee before September 2021, and with permission of the Unit's data manager since then. As such, data can be shared with the permission of the Unit's data manager, Sam Watson s.i.watson@bham.ac.uk.

**Funding:** This research was funded by the National Institute for Health Research (NIHR) Global Health Research Unit on Improving Health in Slums using UK aid from the UK Government to support global health research. The funders had no role in study design, data collection and analysis, decision to publish, or preparation of the manuscript. The views expressed in this publication are those of the author(s) and not necessarily those of the NIHR or the UK Department of Health and Social Care.

**Competing interests:** The authors have declared that no competing interests exist.

and development and an active and healthy life'[1]. Although the right to food is a long-established human right [2, 3] and there is currently enough food produced to supply the entire world's population [4], the FAO estimates that globally over two billion people are food insecure which is disproportionally affecting low- and middle income countries [5, 6]. Going regularly without enough food can have serious consequences for physical and mental health [7]. Next to under- and malnutrition [8], it has been associated with obesity, particularly in adult women, chronic diseases such as type 2 diabetes [9, 10], and social issues including engagement in criminal activities or problems with substance abuse and housing [7, 11].

The United Nations Human Settlements Programme (UN-Habitat) define slums as 'any specific place, whether a whole city, or a neighbourhood, [. . .] if half or more of all households lack improved water, improved sanitation, sufficient living area, durable housing, secure tenure, or combinations thereof' [12]. They estimate that currently more than one billion people worldwide live in slums and project a rise up to five billion by 2030 if current trends continue [13]. Approximately 881 million of these slum dwellers reside in low- and middle-income countries, in particular in Sub Saharan Africa and South-Asia [14].

Even though a large share of the world's population lives in slums and food insecurity is considered a severe health hazard, slum inhabitants are an understudied population. The relatively few previous studies indicate a high prevalence of food insecurity in several slums across Africa and Southeast Asia [7, 15, 16]. However, existing knowledge on the burden and determinants of food insecurity in these settings is limited, potentially outdated, and existing data is not comparable between slum sites.

As depicted by our conceptual framework in Fig 1, which is adapted from a framework published by the International Food Policy Research Institute (IFPRI) in 2006 [17], the

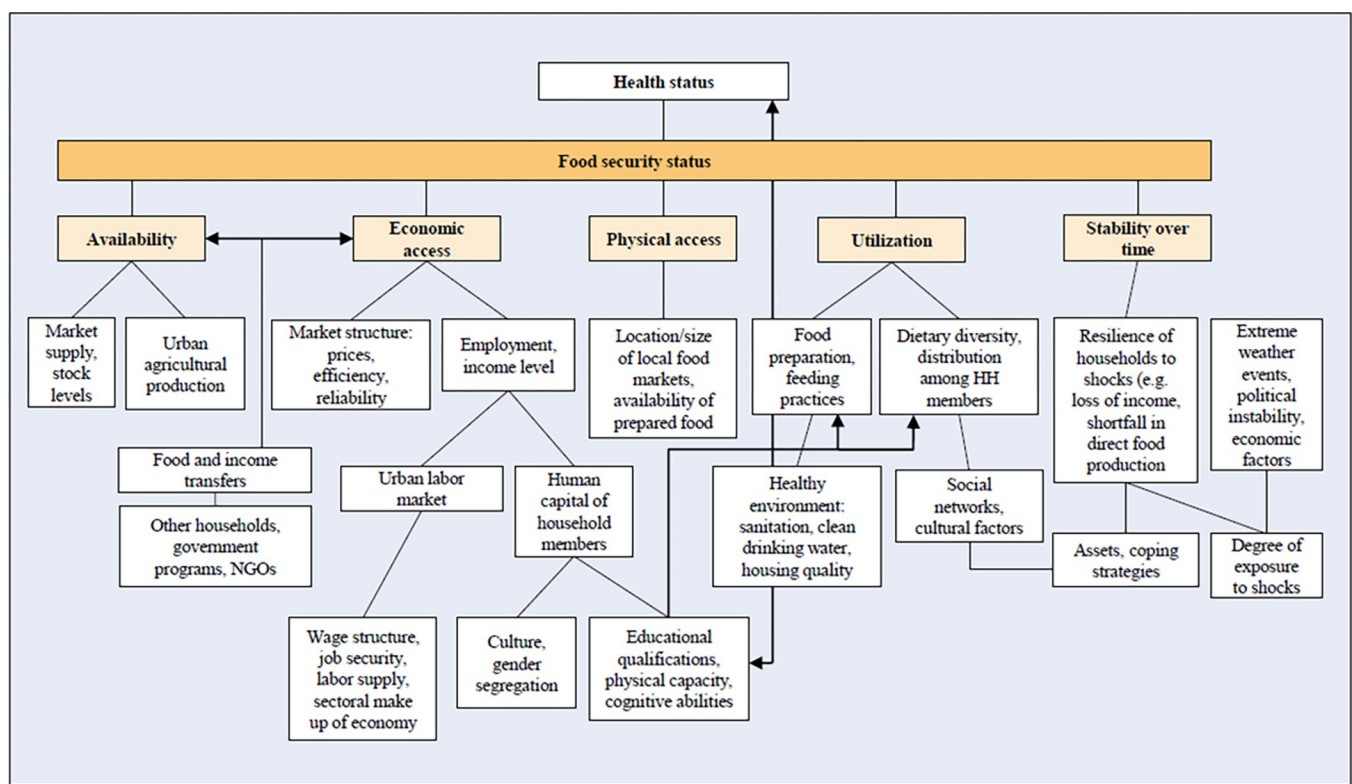

**Fig 1. Conceptual framework of determinants of household food security.**

potential underlying mechanisms causing food insecurity in households are complex. According to the FAO, four dimensions have to be fulfilled to render a household food secure [18], and these four dimensions were used to re-structure the IFPRI framework. Slum residence has clear links to many of the pathways to food (in)security illustrated in Fig 1. For example, educational, cultural and employment factors can affect human capital of slum household members, linked to poorer economic access to food. As an additional example, many slums are likely to be particularly vulnerable to extreme weather events, compared to other neighbourhoods [19], which is linked to instability across the other key dimensions for food security (availability, economic and physical access, utilisation).

Previous studies establish economic resources as an important factor for food security in slums [15, 17, 20–25], with educational factors also associated with food security [7, 16, 20–23, 25] although not in all studies [17]. Single studies suggest that the gender of the household head and the number of adult women in a household affect food insecurity [21, 25]. Other factors are little researched, such as the relationship between having a kitchen or certain household amenities and being food secure, which would be relevant to investigate in the context of high-dependence on ready-made foods in slums [26–28]. There are contradictory findings relating to some factors, such as migration history [15, 17, 20, 21].

Particularly little research has been conducted on slum sites in Nigeria, we only identified one empirical study from Ibadan [20]. Similarly, we only identified prevalence reports and one case study from Pakistan [29–31]. Further insight into this topic area is crucial for planning effective interventions that target the most vulnerable subgroups of slum inhabitants and direct further research and policy efforts. Using multisite household survey data, this study sought to investigate and compare the prevalence and socio-demographic associations of household food insecurity in seven slum sites in Africa and Asia across the five cities of Lagos and Ibadan, Nigeria, Nairobi, Kenya, Karachi, Pakistan, and Dhaka, Bangladesh.

## Methods

### Data and study setting

Secondary analysis was conducted of anonymised data collected by the NIHR Global Health Research Unit on Improving Health in Slums (https://warwick.ac.uk/fac/sci/med/about/centres/wcfgh/slums). These are large-scale, household, individual and healthcare facility surveys across multiple slum sites in four countries with the primary aim to analyse the health care access and use of slum inhabitants. As study sites, seven slum sites from the five cities Lagos and Ibadan (Nigeria: NG1-3), Nairobi (Kenya: KE1-2), Karachi (Pakistan: PK1), and Dhaka (Bangladesh: BD1) were included.

Full ethical approval was granted by relevant research review boards (Ministry of Health, Lagos State Government (LSMH/2695/11/259); Ministry of Health, Oyo State Government (ADB/479/657); Amref Health Africa (AMREF-ESRC P440/2018); National Bioethics Committee Pakistan (4-87/NBC-298/18/RDC3530); Bangladesh Medical Research Council; University of Warwick Biomedical and Scientific Research Ethics Sub-Committee (REGO-2017-2043 AM01) [32]. All participants provided written informed consent. Ethical approval allowed for anonymised data to be shared with researchers outside the specified Global Health Research Unit with the permission of the Unit's management and financial committee before September 2021, and with permission of the Unit's data manager since then.

### Study design

The methodology of the survey design is described in more detail elsewhere [33]. In summary, a household-based, spatially referenced, cross-sectional, retrospective survey was conducted in

the seven selected slum sites. Sites were chosen by the research teams for pragmatic reasons after considering eligible sites that may the UN definition of a slum and were geographically discrete and named neighbourhoods within city boundaries. Using satellite imagery, a spatially referenced sampling frame was obtained for each site containing all geo-located households in the area of interest [33]. People were considered a household if they were currently 'living in the same housing unit or connected premises' [32]. An inhibitory sampling design with close pairs was used to generate a sample of 1200 well-dispersed households for each site, with the aim of recruiting 1000 households per site in total. Participants were surveyed between December 2018 and 2019 in Nigeria, Kenya and Bangladesh and between May 2019 and June 2020 in Pakistan. Up to three attempts to conduct each individual survey were made in case the participant was absent [33].

## Survey instrument and variable selection

Three survey instruments were employed in the project, of which only the household-level survey was used for this study. One questionnaire per household was completed by the head of household, if available, or another adult in the household if necessary. The household questionnaire was developed to capture demographic and socioeconomic information such as household size or income as well as individual variables including age, sex, education, and migration status of all members [32]. Existing questions, particularly from the Demographic and Health Surveys country-specific questionnaires were adopted where appropriate within the household survey, as well as questions from the World Health Organisation Study of global AGEing and adult health.

The outcome of interest was household food insecurity and was categorized as a binary variable (Yes/No) captured with the following question: *In the past seven days were there any days the household did not have enough food or money to buy food*?. This primarily evaluates two dimensions of food security, namely the availability and economical access to food, however, does not depict how the food is utilized (e.g. dietary diversity). This question was previously used in the 2014 Kenya Demographic and Health Survey questionnaire [34].

The selection of independent variables was based on the findings from the published literature and guided by the conceptual framework mentioned in the introduction (Fig 1). Variables included the highest level of school attended by the household head (Never attended; Primary or middle; Secondary or tertiary), whether the household head is currently working, the age of the household head in years (≤31; 32–39; 40–59; 50–59; 60+), the migration status of the household head (always lived in neighbourhood or not), the sex of the household head (Female/Male), household size (total number of household members), number of children under 12 years in the household, proportion of adult female household members (%), the national wealth quintile (Bottom/Lower/Middle; Upper/Top), the possession of agricultural land by any of the household members (Yes/No), whether the household owns a refrigerator, whether the household does anything to the water to make it safer to drink, and whether the household owns a separate kitchen.

This study restricts analyses only to those households with an identified household head in order to allow for the examination of variables relating to the socio-economic status of the household head.

## Statistical analysis

Frequencies and percentages of household food insecurity were tabulated by covariates and site. Univariable logistic regression analysis was conducted to examine the individual associations of the independent variables with the outcome. Multivariable logistic regression was then

used to identify associations with household food insecurity across the seven slum sites. All known determinants of household food insecurity present in the data set have been included as independent variables and further potential predictors with weak or controversial evidence have been considered such as migration status and the possession of agricultural land. The final decision was made after examining the candidate variables in terms of missing data, sufficient sample sizes, collinearity, and model fit. The variable 'Cash transfer or social assistance' was excluded from the final model because category sizes were considered too small for analysis (only 3·4% of households received cash transfer or social assistance in the pooled sample). Further, the number of people per room used for sleeping was not included due to collinearity concerns (S2 Table).

First, a pooled model was run including data from all seven sites, which was adjusted for slum site (Seven dummy variables: NG1-3; KE1-2; PK1; BD1). Then, the same logistic regression model was run for each of the seven slum sites, separately, adjusted for all of the included independent variables. Model fit was assessed by generating the Hosmer-Lemeshow, Pearson Chi-squared and MC Fadden R-squared statistic which indicated a good fit and predictive power for the pooled model (S3 Table). All statistical analyses were performed using STATA 15.1.

### Patient and public involvement

Patients and the public were not involved in the design, conduct or reporting plans for this secondary analysis.

## Results

### Sample characteristics

There were 7002 households in the full sample with median response rate of 69%, ranging from 94% in BC1 to 57% in KE2.[22] Of the surveyed households 6,546 had a household head identified. Of these 435 were dropped because of missingness and invalid variable values which resulted in a final analytic sample of 6111 households (S1 Table). On average four people were living in one household (range: 3·3–6·2) with about one child below the age of 12 (range: 0·7–1·7). Fifty-four per cent of households were categorized as bottom, lower or middle wealth quintile while 46·5% belonged to the upper and top quintile (Table 1).

The mean age of household heads in the complete sample was 43·8 years, varying from 37·0–52·2 years between slum settings (Table 1). Overall, 80·9% of sampled household heads were male, with 89·2% in site PK1 and 68·3% in site KE1. Across all sites, more than half of the household heads attended secondary or tertiary level education (54·6%), about a third attended primary or middle school (29·2%) and the remaining 16·2% never attended school. Eighty-six per cent of the sampled household heads indicated to be currently working (range: 75·5–91·6%). Other variables differed considerably depending on the study site. Overall, more than half of the household heads had always been living in their neighbourhood (54·3%), however this varied from 2·9 to 94·1% across sites. Forty-one per cent (range: 6·3–88·4%) of households reported having a separate kitchen and 25% (range: 0·9–75·6%) a refrigerator. About one third (range: 6·9–71·4%) reported that any household member owns agricultural land, and a quarter of households prepared their water to make it safer to drink (range: 6·5–44·8%) (Table 1).

### Prevalence of household food insecurity across sites

In total, 2,671 (40·8%) households were identified as food insecure (Table 2). This differed considerably between sites, with the lowest proportions of food insecurity in site BD1 (9·1%)

**Table 1. Baseline household characteristics, Frequency (%)/Mean (Standard deviation).**

| Variable | Nigeria | | | Kenya | | Pak· | Bangl· | Total |
|---|---|---|---|---|---|---|---|---|
| | NG1 | NG2 | NG3 | KE1 | KE2 | PK1 | BD1 | |
| Sample size | 1,070 (16·4) | 722 (11·0) | 730 (11·2) | 1,000 (15·3) | 1,076 (16·4) | 932 (14·2) | 1,016 (15·5) | 6,546 |
| *Demographic characteristics* | | | | | | | | |
| Highest level of school attended (household head) | | | | | | | | |
| Never attended school | 190 (19·7) | 127 (19·0) | 32 (4·6) | 87 (8·8) | 7 (0·7) | 246 (26·7) | 330 (32·6) | 1,019 (16·2) |
| Primary/Middle | 160 (16·6) | 179 (26·8) | 74 (10·7) | 565 (57·0) | 402 (38·3) | 57 (6·2) | 407 (40·2) | 1,844 (29·2) |
| Secondary/Tertiary | 617 (63·8) | 363 (54·3) | 589 (84·8) | 340 (34·3) | 641 (61·1) | 617 (67·1) | 276 (27·3) | 3,442 (54·6) |
| Wealth quintile | | | | | | | | |
| Upper/Top | 526 (49·2) | 282 (39·1) | 540 (74·0) | 268 (26·8) | 506 (47·0) | 142 (15·2) | 778 (76·6) | 3,042 (46·5) |
| Currently working | | | | | | | | |
| Yes | 871 (82·2) | 604 (86·8) | 656 (91·6) | 851 (85·3) | 973 (90·8) | 704 (75·5) | 922 (90·8) | 5,581 (86·0) |
| Age of the household head (years) | | | | | | | | |
| ≤31 | 240 (22·5) | 77 (10·7) | 72 (9·9) | 287 (28·7) | 400 (37·2) | 96 (10·3) | 346 (34·1) | 1,518 (23·2) |
| 32–39 | 151 (14·1) | 74 (10·3) | 153 (21·0) | 209 (21·0) | 265 (24·6) | 152 (16·3) | 249 (24·5) | 1,253 (19·1) |
| 40–59 | 24 (22·7) | 173 (24·0) | 210 (28·8) | 219 (21·9) | 244 (22·7) | 240 (25·8) | 222 (21·9) | 1,551 (23·7) |
| 50–59 | 179 (16·7) | 149 (20·6) | 167 (22·9) | 160 (16·0) | 119 (11·1) | 212 (22·8) | 111 (10·9) | 1,097 (16·8) |
| 60+ | 259 (24·2) | 249 (34·5) | 128 (17·5) | 125 (12·5) | 48 (4·5) | 232 (24·9) | 88 (8·7) | 1,126 (17·2) |
| Sex of the household head | | | | | | | | |
| Male | 896 (83·7) | 561 (77·7) | 634 (86·9) | 683 (68·3) | 874 (81·2) | 831 (89·2) | 819 (80·6) | 5,298 (80·9) |
| Mean household size (SD) | 3·7 (1·9) | 3·5 (1·9) | 4·3 (2·0) | 3·3 (2·23) | 2·5 (1·6) | 6·2 (3·3) | 4·0 (2·0) | 3·9 (2·4) |
| Mean number of children under 12 in the household (SD) | 1·2 (1·3) | 0·9 (1·1) | 1·2 (1·3) | 1·1 (1·3) | 0·7 (1·0) | 1·7 (1·8) | 1·0 (1·1) | 1·1 (1·3) |
| Percentage of adult female household members, Mean (SD) | 32·7 (26·1) | 35·7 (27·5) | 31· 7 (20·3) | 28·1 (26·7) | 25·9 (28·0) | 30·5 (18·0) | 33·3 (19·2) | 30·9 (24·3) |
| Migration status of the household head | | | | | | | | |
| Always lived there | 957 (90·4) | 654 (94·1) | 643 (89·9) | 357 (35·8) | 127 (11·9) | 749 (80·4) | 28 (2·8) | 3,515 (54·3) |
| *Environmental characteristics* | | | | | | | | |
| Separate kitchen | | | | | | | | |
| Yes | 404 (37·8) | 211 (29·3) | 521 (71·4) | 62 (6·9) | 66 (6·3) | 827 (88·7) | 534 (54·0) | 2,625 (41·1) |
| Possession of agricultural land | | | | | | | | |
| Yes | 181 (17·2) | 105 (14·6) | 97 (13·5) | 277 (27·7) | 490 (45·6) | 64 (6·9) | 727 (71·6) | 1,941 (29·8) |
| Refrigerator | | | | | | | | |
| Yes | 299 (27·9) | 140 (19·4) | 404 (55·3) | 54 (5·4) | 28 (2·6) | 705 (75·6) | 9 (0·9) | 1,639 (25·1) |
| Is anything done to the water to make it safer to drink? | | | | | | | | |
| Yes | 69 (6·5) | 58 (8·0) | 69 (9·5) | 180 (18·0) | 482 (44·8) | 393 (42·3) | 428 (42·1) | 1,679 (25·7) |

Figures displayed in this table are frequencies (%) except when indicated as mean (SD: Standard deviation)

**Table 2. Prevalence of food insecurity across household characteristics (Frequency (%)/Mean (Standard deviation)) and univariate logistic regression analysis (Odds ratio (95% confidence interval)).**

| Variable | Household food insecurity | | OR (95% CI) | Total |
|---|---|---|---|---|
| | **No** | **Yes** | | |
| **Total** | **3,874 (59·2)** | **2,671 (40·8)** | - | **6,545** |
| Site | | | - | |
| NG1 | 385 (36·0) | 685 (64·0) | | 1,070 |
| NG2 | 225 (31·2) | 496 (68·8) | | 721 |
| NG3 | 281 (38·5) | 449 (61·5) | | 730 |
| KE1 | 590 (59·0) | 410 (41·0) | | 1,000 |
| KE2 | 920 (85·5) | 156 (14·5) | | 1,076 |
| PK1 | 549 (58·9) | 383 (41·1) | | 932 |
| BD1 | 924 (90·9) | 92 (9·1) | | 1,016 |
| Highest level of school attended (household head) | | | | |
| Never attended school | 632 (62·0) | 387 (38.0) | Ref· | 1,019 |
| Primary/Middle | 1,151 (62·4) | 693 (37·6) | 0·98 (0·84–1·15) | 1,844 |
| Secondary/Tertiary | 1,993 (57·9) | 1,450 (42·1) | 1·19* (1·03–1·37) | 3,443 |
| Wealth quintile | | | | |
| Bottom/Lower/Middle | 1,879 (53·6) | 1,624 (46·4) | Ref· | 3,503 |
| Upper/Top | 1,995 (65·6) | 1,047 (34·4) | 0·61* (0·55–0·67) | 3,042 |
| Currently working | | | | |
| No | 493 (54·2) | 416 (45·8) | Ref· | 909 |
| Yes | 3,358 (60·2) | 2,223 (39·8) | 0·78* (0·68–0·90) | 5,581 |
| Age of the household head (years) | | | | |
| ≤31 | 1,055 (69·5) | 463 (30·5) | Ref· | 1,518 |
| 32–39 | 777 (62·0) | 476 (38·0) | 1·40* (1·19–1·63) | 1,253 |
| 40–59 | 879 (56·7) | 671 (43·3) | 1·74* (1·50–2·02) | 1,550 |
| 50–59 | 591 (53·87) | 506 (46·1) | 1·95* (1·66–2·29) | 1,097 |
| 60+ | 571 (50·7) | 555 (49·3) | 2·21* (1·89–2·60) | 1,126 |
| Sex of the household head | | | | |
| Female | 730 (58·5) | 518 (41·5) | Ref· | 1,248 |
| Male | 3,144 (59·4) | 2,153 (40·7) | 0·97 (0·85–1·09) | 5,287 |
| Mean household size (SD) | 3·8 (2·4) | 4·1 (2·4) | 1·05* (1·03–1·08) | 3·9 (2·4) |
| Mean number of children under 12 in the household (SD) | 1·0 (1·3) | 1·2 (1·4) | 1·11* (1·07–1·15) | 1·1 (1·3) |
| Percentage of adult female household members, Mean (SD) | 30·8 (24·6) | 30·9 (24·0) | 1·00 (1·00–1·00) | 30·9 (24·3) |
| Migration status of the household head | | | | |
| Not always lived in this neighbourhood | 2,265 (76·4) | 699 (23·6) | Ref· | 2,964 |
| Always lived in this neighbourhood | 1,576 (44·8) | 1,939 (55·2) | 3·99* (3·58–4·44) | 3,515 |
| Separate kitchen | | | | |
| No | 2,247 (59·7) | 1,518 (40·3) | Ref· | 3,765 |
| Yes | 1,519 (57·9) | 1,106 (42·1) | 1·08 (0·97–1·19) | 2,625 |
| Possession of agricultural land | | | | |
| No | 2,399 (52·5) | 2,170 (47·5) | Ref· | 4,569 |
| Yes | 1,458 (75·1) | 483 (24·9) | 0·37* (0·33–0·41) | 1,941 |
| Refrigerator | | | | |
| No | 2,992 (61·0) | 1,915 (39·0) | Ref· | 4,907 |
| Yes | 882 (53·9) | 756 (46·2) | 1·33* (1·20–1·50) | 1,638 |
| Is anything done to the water to make it safer to drink? | | | | |
| No | 2,677 (55·0) | 2,187 (45·0) | Ref· | 4,864 |

*(Continued)*

**Table 2.** (Continued)

| Variable | Household food insecurity | | OR (95% CI) | Total |
|---|---|---|---|---|
| | **No** | **Yes** | | |
| Yes | 1,195 (71·2) | 484 (28·8) | 0·50* (0·44–0·56) | 1,679 |

Figures displayed in this table are frequencies (%) except when indicated as mean (SD: Standard deviation)

Ref·: Reference category

* indicates significance at the 5% level

and KE2 (14·5%) and the highest burden observed in site NG2 (68·8%) followed by NG1 (64%).

## Model based socio-demographic associations of household food insecurity across and within sites

In the pooled model (Table 3), households from the upper or top wealth quintile were about 35% less likely to be food insecure than those from lower quintiles (OR: 0·65; CI: 0·57–0·75). Similarly, where household heads were currently working, compared to not working, there were 18% lower odds of household food insecurity (OR: 0·82; CI: 0·69–0·98). The odds of households being food insecure were 26% higher for households with heads aged between 32 to 39 years (OR: 1·26; CI: 1·04–1·53) and 23% for those 40 to 49 years (OR: 1·23; CI: 1·02–1·48) compared to household heads aged 31 years and younger. Where the household head attended primary or middle school, households were about 50% more likely to be food insecure compared to those who never attended school (OR: 1·52; CI: 1·24–1·86). Those where household heads had secondary education or higher, however, did not significantly differ statistically in their odds ratio from those without education. In cases where any of the household members owned agricultural land, households were 20% (OR: 0·80; CI: 0·69–0·94) and where they owned a refrigerator 40% (OR: 0·60; CI: 0·51–0·70) less likely to be food insecure.

In the site-specific analysis (Table 4), the odds of being food insecure were also higher for both education levels in sites NG1, NG3, KE2 compared to household heads never attending school. This was, however, only statistically significant in site NG1, where those with formal education were between 2·4–2·8 times more likely to suffer from household food insecurity (OR: 2·4; CI: 1·47–3·81; OR: 2·80; CI: 1·85–4·20). On the contrary, in slum NG2 for both primary and middle school and secondary or higher education likeliness of household food insecurity was reduced. Households were less likely to be insecure when their household head attended secondary or tertiary education in the remaining three sites being statistically significant in PK1 (OR: 0·69) and BD1 (OR: 0·51). For households from the upper or top national wealth quintile there was a significant reduction in the odds of being food insecure by 55% in BD1 (OR: 0·47; CI: 0·28–0·78), 45% in NG1 (OR: 0·55; CI: 0·41–0·76), and 47% in KE1 (OR: 0·53; CI: 0·38–0·74). The odds of being food insecure for households with working household heads were consistently reduced across sites with the biggest reduction of 65% recorded in NG3 (OR: 0·35). In site NG1 for each additional household member, the odds of food insecurity increased by 29% (OR: 1·29; CI: 1·12–1·49) and in NG2 by 18% (OR: 1·18; CI: 1·10–1·37). The impact of the household head's migration status on food insecurity varied between sites but was only statistically significant in NG2. In this site, where household heads have always been living in the neighbourhood, households had more than three times the odds of food insecurity (OR: 3·20; CI: 1·58–6·42) compared with households where heads are migrants to the slum.

**Table 3. Logistic regression analysis for household food insecurity (pooled for all slum sites).**

| Independent variable | OR | P-value | 95% CI |
|---|---|---|---|
| Site | | | |
| NG1 | Ref· | | |
| NG2 | 1·18 | 0·145 | 0·95 to 1·46 |
| NG3 | 1·15 | 0·199 | 0·93 to 1·44 |
| KE1 | 0·28 | 0·000 | 0·22 to 0·36 |
| KE2 | 0·08 | 0·000 | 0·06 to 0·11 |
| PK1 | 0·39 | 0·000 | 0·31 to 0·50 |
| BD1 | 0·06 | 0·000 | 0·04 to 0·08 |
| Highest level of school attended (household head) | | | |
| Never attended school | Ref· | | |
| Primary/Middle | 1·52 | <0·000 | 1·24 to 1·86 |
| Secondary/Tertiary | 1·11 | 0·275 | 0·92 to 1·34 |
| Wealth quintile | | | |
| Bottom/Lower/Middle | Ref· | | |
| Upper/Top | 0·65 | <0·000 | 1·90 to 2·38 |
| Currently working | | | |
| No | Ref· | | |
| Yes | 0·82 | 0·030 | 0·69 to 0·98 |
| Age of the household head (years) | | | |
| ≤31 | Ref· | | |
| 32–39 | 1·26 | 0·019 | 1·04 to 1·53 |
| 40–59 | 1·23 | 0·030 | 1·02 to 1·48 |
| 50–59 | 1·19 | 0·103 | 0·97 to 1·47 |
| 60+ | 1·02 | 0·882 | 0·82 to 1·27 |
| Sex of the household head | | | |
| Female | Ref· | | |
| Male | 0·86 | 0·111 | 0·71 to 1·04 |
| Household size | 1·01 | 0·502 | 0·97 to 1·06 |
| Number of children under 12 in the household | 1·04 | 0·237 | 0·97 to 1·12 |
| Percentage of adult female household members (mean) | 0·996 | 0·022 | 0·993 to 0·999 |
| Migration status of the household head | | | |
| Not always lived there | Ref· | | |
| Always lived there | 1·06 | 0·482 | 0·90 to 1·23 |
| Separate kitchen | | | |
| No | Ref· | | |
| Yes | 1·01 | 0·880 | 0·87 to 1·18 |
| Possession of agricultural land | | | |
| No | Ref· | | |
| Yes | 0·80 | 0·005 | 0·69 to 0·94 |
| Refrigerator | | | |
| No | Ref· | | |
| Yes | 0·60 | <0·000 | 0·51 to 0·70 |
| Is anything done to the water to make it safer to drink? | | | |
| No | Ref· | | |
| Yes | 1·09 | 0·251 | 0·94 to 1·27 |

Logistic regression model adjusted for site.

Ref·: Reference category

**Table 4. Logistic regression analysis for household food insecurity (separate for each slum site).**

| Independent variable | OR (95% Confidence Interval) | | | | | | |
|---|---|---|---|---|---|---|---|
| | Nigeria | | | Kenya | | Pakistan | Bangl· |
| | NG1 | NG2 | NG3 | KE1 | KE2 | PK1 | BD1 |
| Highest level of school attended (household head) | | | | | | | |
| Never attended school | Ref· | Ref· | Ref· | Ref· | Ref· | Ref· | Ref· |
| Primary/Middle | 2·36 (1·47–3·81)** | 0·93 (0·51–1·67) | 1·08 (0·41–2·88) | 1·21 (0·71–2·06) | 1·83 (0·20–16·87) | 1·13 (0·63–2·05) | 1·17 (0·68–2·00) |
| Secondary/Tertiary | 2·79 (1·85–4·20)** | 0·57 (0·32–1·05) | 1·02 (0·43–2·44) | 0·81 (0·45–1·44) | 1·32 (0·14–12·20) | 0·69 (0·50–0·96)* | 0·51 (0·25–1·03) |
| Wealth quintile | | | | | | | |
| Bottom/Lower/Middle | Ref· | Ref· | Ref· | Ref· | Ref· | Ref· | Ref· |
| Upper/Top | 0·55 (0·41–0·76)** | 1·06 (0·72–1·56) | 0·78 (0·50–1·23) | 0·53 (0·38–0·74)** | 0·79 (0·54–1·14) | 1·13 (0·76–1·68) | 0·47 (0·28–0·78)** |
| Currently working | | | | | | | |
| Yes | Ref· | Ref· | Ref· | Ref· | Ref· | Ref· | Ref· |
| No | 1·26 (0·84–1·87) | 1·21 (0·71–2·07) | 0·35 (0·17–0·73)** | 0·55 (0·36–0·85)** | 0·36 (0·21–0·61)** | 1·19 (0·81–1·75) | 0·49 (0·22–1·10) |
| Age of the household head (years) | | | | | | | |
| ≤31 | Ref· | Ref· | Ref· | Ref· | Ref· | Ref· | Ref· |
| 32–39 | 1·11 (0·67–1·85) | 0·78 (036–1·70) | 1·02 (0·53–1·96) | 1·33 (0·87–2·04) | 1·20 (0·73–1·99) | 1·36 (0·79–2·35) | 1·61 (0·87–2·99) |
| 40–59 | 1·06 (0·65–1·73) | 0·87 (0·44–1·73) | 0·97 (0·51–1·84) | 1·32 (0·85–2·04) | 1·23 (0·73–2·07) | 1·46 (0·88–2·44) | 0·83 (0·40–1·73) |
| 50–59 | 0·91 (0·53–1·55) | 0·91 (0·45–1·84) | 0·94 (0·47–1·88) | 1·46 (0·90–2·36) | 1·05 (0·54–2·05) | 1·16 (0·67–2·02) | 1·21 (0·51–2·85) |
| 60+ | 0·93 (0·55–1·55) | 0·66 (0·33–1·34) | 0·74 (0·34–1·60) | 1·31 (0·76–2·24) | 1·50 (0·67–3·38) | 1·19 (0·67–2·12) | 0·45 (0·14–1·41) |
| Sex of the household head | | | | | | | |
| Female | Ref· | Ref· | Ref· | Ref· | Ref· | Ref· | Ref· |
| Male | 0·66 (0·38–1·13) | 1·41 (0·80–2·51) | 0·68 (0·38–1·23) | 0·70 (0·48–1·03) | 0·80 (0·43–1·49) | 1·03 (0·64–1·02) | 0·97 (0·47–1·98) |
| Household size | 1·29 (1·12–1·49)** | 1·18 (1·01–1·37)* | 1·02 (0·90–1·17) | 0·93 (0·83–1·03) | 0·90 (0·71–1·14) | 0·95 (0·65–1·33) | 1·04 (0·86–1·26) |
| Number of children under 12 in the household | 0·78 (0·64–0·95) | 1·07 (0·83–1·39) | 1·13 (0·92–1·38) | 1·18 (0·96–1·44) | 1·13 (0·77–1·66) | 1·11 (0·97–1·26) | 0·89 (0·63–1·24) |
| Percentage of adult female household members (mean) | 1·00 (0·99–1·01) | 1·00 (0·99–1·01) | 0·99 (0·98–1·00) | 0·99 (0·98–1·00)** | 1·00 (0·99–1·01) | 1·01 (1·00–1·01) | 0·98 (0·97–1·00)* |
| Migration status of the household head | | | | | | | |
| Not always lived there | Ref· | Ref· | Ref· | Ref· | Ref· | Ref· | Ref· |
| Always lived there | 1·47 (0·89–2·41) | 3·19 (1·58–6·42)** | 1·63 (0·93–2·88) | 0·81 (0·59–1·14) | 0·80 (0·43–1·49) | 0·93 (0·65–1·33) | 0·97 (0·20–4·57) |
| Separate kitchen | | | | | | | |
| No | Ref· | Ref· | Ref· | Ref· | Ref· | Ref· | Ref· |
| Yes | 0·61 (0·45–0·83)** | 0·75 (0·50–1·12) | 0·68 (0·45–1·04) | 0·77 (0·41–1·42) | 0·94 (0·42–2·11) | 2·34 (1·43–3·83)** | 5·12 (2·87–9·13)** |
| Possession of agricultural land | | | | | | | |
| No | Ref· | Ref· | Ref· | Ref· | Ref· | Ref· | Ref· |
| Yes | 0·89 (0·61–1·32) | 1·16 (0·69–1·96) | 0·82 (0·51–1·33) | 0·51 (0·37–0·72)** | 1·72 (1·15–2·56)** | 0·55 (0·31–0·98)* | 0·46 (0·29–0·75)** |
| Refrigerator | | | | | | | |
| No | Ref· | Ref· | Ref· | Ref· | | Ref· | Ref· |

*(Continued)*

**Table 4.** (Continued)

| Independent variable | OR (95% Confidence Interval) | | | | | | |
|---|---|---|---|---|---|---|---|
| | Nigeria | | | Kenya | | Pakistan | Bangl· |
| | NG1 | NG2 | NG3 | KE1 | KE2 | PK1 | BD1 |
| Yes | 0·64 (0·46–0·89)* | 0·46 (0·30–0·72)** | 0·44 (0·31–0·63)** | 0·18 (0·07–0·47)** | - | 0·55 (0·68–1·28) | 1·78 (0·20–1·62) |
| Is anything done to the water to make it safer to drink? | | | | | | | |
| No | Ref· | Ref· | Ref· | Ref· | Ref· | Ref· | Ref· |
| Yes | 1·34 (0·73–2·46) | 2·18 (1·01–4·42)* | 1·53 (0·86–2·75) | 1·35 (0·94–1·95) | 0·82 (0·55–1·20) | 1·03 (0·77–1·37) | 1·06 (0·66–1·71) |

Logistic regression model adjusted for all factors listed in the table.

Ref·: Reference category

**p<0·01

*p<0·05

The directions of the association between having a separate kitchen and food insecurity status were mixed between sites and were statistically significant in sites NG1 (OR: 0·6; CI: 0·45–0·83), PK1 (OR: 2·34; CI: 1·43–3·83) and BD1 (OR: 5·12; CI: 2·87–9·13). The possession of land was favourable for food security in most site-specific models, with significant reductions in the odds of food insecurity in KE1 (OR: 0·51; CI: 0·37–0·72), PK1 (OR: 0·55; CI: 0·31–0·98) and BD1 (OR: 5·12; CI: 2·87–9·13). Only in site KE2, households which owned land tended to be food insecure more often (OR: 1·72; CI: 1·15–2·56). The association between the possession of a refrigerator and food insecurity was statistically significant in sites NG1 (OR: 0·64; CI: 0·46–0·90), NG2 (OR: 0·46; CI: 0·30–0·72), NG3 (OR: 0·44; CI: 0·31–0·63), and KE1 (OR: 0·18; CI: 0·07–0·47). No reliable results could be obtained for site KE2 due to only 28 households (2·60%) having a refrigerator. Contrary to the statistically insignificant outcomes for the other slums, in NG2, the odds of food insecurity were more than doubled for those households which did something to make their water safer to drink, compared to those which did not (OR: 2·18; CI: 1·08–4·42).

## Discussion

This study identified that about 40% of households residing in slums were food insecure which varied between the seven slum sites. Economic resources represented by the working status of the household head and the national wealth quintile were also important for household food insecurity. Belonging to the upper and top wealth quintiles and the employment of the household head were both linked to lower levels of household insecurity, which was mostly consistent across slum sites. Those households who owned a refrigerator or agricultural land were 40% or 20% less likely to report food insecurity, respectively with one exception regarding agricultural land, in a slum in Nairobi, Kenya. Migration status of the household head was not associated with household food insecurity status in the pooled sample but in one site in Nigeria the likelihood of food insecurity increased by more than three times if household heads were not migrants.

This study showed that prevalence of food insecurity differs greatly between slum sites and was often lower than previous study findings on similar areas. This may be due to the specific question used in the survey. The burden of food insecurity was the highest in Nigeria. In the two slums in Ibadan, 64·2 and 68·8% of households were food insecure which is lower than what was reported for a slum site in Ibadan in 2018, where 81% of households were categorized

as food insecure [20]. The extent of food insecurity deviated in the two sites in Kenya (KE1: 41%; KE2: 15·5%) and was slightly lower than a range of estimates from previous studies in slums sites in Kenya between 2011–2016 which reported a prevalence of 35–49% [7, 15, 16]. The extent of household food insecurity in the slum in Karachi, Pakistan (41·1%) was lower han what was found for a slum population in the same city in 2019, with two-thirds of households being food insecure [29]. With 9·1%, the prevalence in Dhaka, Bangladesh, was remarkably lower than the results of a recent 2021 study (51·7%), defining those households as food insecure who did not consume the recommended minimum nutritional requirements [21].

The causal factors of household food insecurity are complex and possible pathways can be categorized according to four different dimensions that are to be fulfilled simultaneously to ensure food security (Fig 1). A range of studies in slums found strong associations between economic resources such as income, expenditure, debt and employment, and household food insecurity [15, 17, 20–25]. The results of this study for the working status of the household head and households' wealth quintile also suggest that the available financial resources directly influence the economic access to sufficient food. Households, where the household head is employed, may be more likely to secure income for out-of-pocket expenditures on food. Similarly, belonging to a higher wealth quintile, a more general measure of the living standard and long-term financial condition of households, favours availability and access to food resources. Surprisingly, only few households were in the bottom and lower wealth quintile across the slum sites which might be because wealth quintiles are determined at national level. Despite fulfilling the criteria to be considered slums, the included sites were in economic centres of their respective countries and thus participating households are possibly better off in terms of wealth than residents of rural areas or less well-connected urban areas.

Overall migration status of the household head was not significantly associated with the food security status of slum households. This confirms previous studies on a slum in Nairobi in 2014 [15] and multiple slum areas in Bangladesh from 2006 [17] which found no relation between the duration of stay or migration history and food insecurity. A strong positive association was, however, found in one site in Ibadan, Nigeria (NG2) which contradicts Obayelu et al. [20] who reported that the food insecurity prevalence was increasing with decreasing years of stay in a slum Ibadan, possibly explained by better integration into the social network providing better informal insurance and the development of more effective coping strategies. In NG2, similar to the other slum sites in Nigeria, the large majority of household heads have always lived there (94·1%), indicating that there might be less population turnover compared to other slum sites. But because the sample size is very small for migrants in NG2 the finding must also be interpreted with caution. Why new residents to this slum site seem to be better equipped against food insecurity than those native to this site requires further investigation.

About a third of households from the pooled sample reported owning agricultural land and this was linked to reduced household food insecurity. Access to farmland facilitates the availability of food by allowing households to produce crops for self-consumption. This supports Crush et al. [35] who stress the importance of urban agriculture for mitigating food insecurity and dependence on the food market among urban poor households, in particular in Africa. Previous study findings from Bangladesh [17], in contrast, did not reveal a significant association between land ownership and food security status and highlight that the degree to which agriculture for home consumption is contributing to food security also depends on what opportunity costs are created, such as the time that is not spent engaging in the labour market. This might partially explain the exception seen in KE2 showing increased odds of food insecurity for those with farmland. Another interfering factor might also have been adverse weather conditions at that time which disadvantaged households who partially rely on subsistence farming as a source of food.

While this study was one of the first multi-site studies on household food insecurity in slum settings using a more representative and large-scale sample compared to previous household surveys there are inevitable limitations. First, there might have been residual selection bias given that the response rate of households varied across the slum sites and the inclusion of households with household heads only. Thus, non-respondents, households without a formal household head, or those excluded from our complete case analysis due to missing data, might have differed from those included. Second, the outcome measure is limited to a single indicator which mainly assesses the economic access dimension of household food insecurity and is subject to the judgement of the surveyed participant and potential recall bias. It has also never been assessed for reliability and validity to our knowledge, but has been used in previous surveys [34]. Third, a few of the included variables have small categories in the site-specific regression models (e.g. refrigerator ownership, education level). Thus, no reliable results could be obtained for some associations in specific sites. Fourth, although we were guided by our conceptual model (Fig 1), we were not able to explore all the potential factors of interest in the dataset, or all the dimensions of food insecurity, as this was secondary analysis of data collected to answer other primary questions. Finally, the cross-sectional nature of the study does not provide information on the causality of the observed relationships, hence, reverse causation cannot be excluded.

Further research is warranted into the role of the other dimensions of food insecurity as well as longitudinal data. The high proportion of food insecure households, particularly in the sites in Nigeria calls for timely slum-focussed policy programmes. As advocated by the FAO [36], a multifaceted policy approach is suggested that combines direct nutritional interventions to relieve acute hunger with long-term, context-specific development investments to improve the livelihoods of slum dwellers.

In conclusion, this study found that the overall extent of food insecurity was high, over 40%, but varied between slum sites. Factors of economic access were found to be important determinants of food insecurity across sites. Other predictors such as migration status and the possession of agricultural land showed varying effects across slum sites even from the same city stressing the need to consider context-specific pathways when formulating strategies against food insecurity. In light of the increasing number of slum inhabitants worldwide, the attention of researchers and policy-makers should be directed at understanding and tackling causes of food insecurity for people living in slums.

## Supporting information

**S1 Table. Missing values and final sample sizes for the included variables.**
(DOCX)

**S2 Table.** A) Other considered household characteristics, Frequency (%)/Mean (Standard deviation). B) Prevalence of food insecurity across other considered household characteristics, Frequency (%)/Mean (Standard deviation).
(DOCX)

**S3 Table. Model fit statistics for the pooled and site-specific logistic regression models.**
(DOCX)

## Acknowledgments

The Improving Health in Slums Collaborative is (in alphabetical order):
     *African Population and Health Research Centre (APHRC), Nairobi, Kenya*

Pauline Bakibinga, Caroline Kabaria, Ziraba Kasiira, Peter Kibe, Catherine Kyobutungi, Nelson Mbaya, Blessing Mberu, Shukri Mohammed, Anne Njeri,

*Aga Khan University, Karachi, Pakistan*

Iqbal Azam, Romaina Iqbal, Ahsana Nazish, Narjis Rizvi,

*Independent University, Bangladesh, Dhaka, Bangladesh*

Syed A. K. Shifat Ahmed, Nazratun Choudhury, Ornob Alam, Afreen Zaman Khan, Omar Rahman, Rita Yusuf

*Nigerian Academy of Sciences, Lagos, Nigeria*

Doyin Odubanjo

*University of Ibadan, Ibadan, Nigeria*

Motunrayo Ayobola, Olufunke Fayehun, Akinyinka Omigbodun, Mary Osuh, Eme Owoaje, Olalekan Taiwo

*University of Birmingham, Birmingham, UK*

Richard J Lilford, Jo Sartori, Samuel I Watson

*University of Glasgow, Glasgow, UK*

João Porto de Albuquerque, Godwin Yeboah

*University of Lancaster, Lancaster, UK*

Peter J Diggle

University of Newcastle, Newcastle, UK

Navneet Aujla,

University of Warwick, Coventry, UK

Yen-Fu Chen, Paramjit Gill, Frances Griffiths, Bronwyn Harris, Jason Madan, Helen Muir, Oyinlola Oyebode, Vangelis Pitidis, Simon Smith, Celia Brown, Philip Ulbrich, Olalekan A Uthman, Ria Wilson, Ji-Eun Park

## Author Contributions

**Conceptualization:** Anthony A. Laverty, Oyinlola Oyebode.

**Formal analysis:** Clara Spieker.

**Supervision:** Anthony A. Laverty, Oyinlola Oyebode.

**Writing – original draft:** Clara Spieker.

**Writing – review & editing:** Anthony A. Laverty, Oyinlola Oyebode.

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
