## [Decision Letter · Decision Letter 0]

31 Aug 2022

PONE-D-22-14578The prevalence and socio-demographic associations of household food insecurity in seven slum sites across Nigeria, Kenya, Pakistan, and Bangladesh. A cross-sectional study.PLOS ONE

Dear Dr. Oyebode,

Thank you for submitting your manuscript to PLOS ONE. After careful consideration, we feel that it has merit but does not fully meet PLOS ONE’s publication criteria as it currently stands. Therefore, we invite you to submit a revised version of the manuscript that addresses the points raised during the review process.

Major Revisions 

We look forward to receiving your revised manuscript.

Kind regards,

Faisal Abbas, PhD

Academic Editor

PLOS ONE

Journal Requirements:

This research was funded by the National Institute for Health Research (NIHR) Global Health Research Unit on Improving Health in Slums using UK aid from the UK Government to support global health research. The views expressed in this publication are those of the author(s) and not necessarily those of the NIHR or the UK Department of Health and Social Care. 

This research was funded by the National Institute for Health Research (NIHR) Global Health Research Unit on Improving Health in Slums using UK aid from the UK Government to support global health research. The views expressed in this publication are those of the author(s) and not necessarily those of the NIHR or the UK Department of Health and Social Care. 

However, funding information should not appear in the Acknowledgments section or other areas of your manuscript. We will only publish funding information present in the Funding Statement section of the online submission form. 

This research was funded by the National Institute for Health Research (NIHR) Global Health Research Unit on Improving Health in Slums using UK aid from the UK Government to support global health research. The views expressed in this publication are those of the author(s) and not necessarily those of the NIHR or the UK Department of Health and Social Care. 

None

6. We note that you have indicated that data from this study are available upon request. PLOS only allows data to be available upon request if there are legal or ethical restrictions on sharing data publicly. For more information on unacceptable data access restrictions, please see http://journals.plos.org/plosone/s/data-availability#loc-unacceptable-data-access-restrictions. 

7. We note that you have referenced (ie. Bewick et al. [5]) which has currently not yet been accepted for publication. Please remove this from your References and amend this to state in the body of your manuscript: (ie “Bewick et al. [Unpublished]”) as detailed online in our guide for authors

Additional Editor Comments:

Major Revisions Suggested

Reviewers' comments:

Reviewer's Responses to Questions

**Comments to the Author**

1. Is the manuscript technically sound, and do the data support the conclusions?

Reviewer #1: Yes

Reviewer #2: Yes

2. Has the statistical analysis been performed appropriately and rigorously? 

Reviewer #1: Yes

Reviewer #2: Yes

3. Have the authors made all data underlying the findings in their manuscript fully available?

Reviewer #1: Yes

Reviewer #2: Yes

4. Is the manuscript presented in an intelligible fashion and written in standard English?

Reviewer #1: Yes

Reviewer #2: Yes

5. Review Comments to the Author

Reviewer #1: The study provides insights on an important topic. However, the paper can be improved by considering the following:

1. The introduction needs to be improved. More relevant literature should be included.

2. The study claims that existing literature is limited in potential as the four dimensions ( depicted in figure 1) suggested by FAO are essential for categorization of food security. The present study also does not incorporate the entire channel mentioned in Figure 1 and many dimensions are ignored. The contribution of this study must be stated clearly and its relevance with Figure 1 be established.

3. In logistic regressions, household size and number of children under the age of 12 are included as independent variables. These variables are collinear in nature and ideally only one of them should be used in the regressions.

4. The regressions also include number of adult females in the household. The concern raised in point 3 is also valid here. The impact of presence of adult females on food insecurity of the household is not stated in the manuscript.

5. Number of dependents in the house including underage children and elderly would be relevant for determining food insecurity.

Reviewer #2: 1. Please provide the background situation of each country in introduction in order to make reader understand the context and be able to compare and contrast.

2. The problem statement is weak as well.

3. Provide ample literature evidence to support the problem.

4. Provide that whether survey instrument was adopted or adapted. Cite the references as well from where the questionnaire was made.

5. The survey design should mention the reliability and validity of the instrument.

6.The sampling design is missing. Inform about the population of the country and then the selected city. Also mention that how many people were surveyed from each city and why. What was the sampling strategy?

6. PLOS authors have the option to publish the peer review history of their article (what does this mean?). If published, this will include your full peer review and any attached files.

Reviewer #1: No

Reviewer #2: No

---

## [Author Response · Author response to Decision Letter 0]

13 Oct 2022

Dear Dr Faisal Abbas, Thank you for the opportunity to respond to the reviewers' comments on our article. I have attached a letter including a point-by-point response to both your editorial queries and the reviewer concerns. With best wishes, Oyinlola Oyebode

---

## [Decision Letter · Decision Letter 1]

24 Nov 2022

The prevalence and socio-demographic associations of household food insecurity in seven slum sites across Nigeria, Kenya, Pakistan, and Bangladesh. A cross-sectional study.

PONE-D-22-14578R1

Dear Dr. Oyebode,

We’re pleased to inform you that your manuscript has been judged scientifically suitable for publication and will be formally accepted for publication once it meets all outstanding technical requirements.

Kind regards,

Faisal Abbas, PhD

Academic Editor

PLOS ONE

Additional Editor Comments (optional):

Accept.

Reviewers' comments:

Reviewer's Responses to Questions

**Comments to the Author**

1. If the authors have adequately addressed your comments raised in a previous round of review and you feel that this manuscript is now acceptable for publication, you may indicate that here to bypass the “Comments to the Author” section, enter your conflict of interest statement in the “Confidential to Editor” section, and submit your "Accept" recommendation.

Reviewer #1: All comments have been addressed

Reviewer #2: All comments have been addressed

2. Is the manuscript technically sound, and do the data support the conclusions?

Reviewer #1: Yes

Reviewer #2: Yes

3. Has the statistical analysis been performed appropriately and rigorously? 

Reviewer #1: Yes

Reviewer #2: Yes

4. Have the authors made all data underlying the findings in their manuscript fully available?

Reviewer #1: Yes

Reviewer #2: Yes

5. Is the manuscript presented in an intelligible fashion and written in standard English?

Reviewer #1: Yes

Reviewer #2: Yes

6. Review Comments to the Author

Reviewer #1: (No Response)

Reviewer #2: The paper has been improved and the comments have been well addressed by the authors. I recommend the paper for the publucation.

7. PLOS authors have the option to publish the peer review history of their article (what does this mean?). If published, this will include your full peer review and any attached files.

Reviewer #1: No

Reviewer #2: No

---

## [Editor Report · Acceptance letter]

21 Dec 2022

PONE-D-22-14578R1 

The prevalence and socio-demographic associations of household food insecurity in seven slum sites across Nigeria, Kenya, Pakistan, and Bangladesh. A cross-sectional study. 

Dear Dr. Oyebode:

I'm pleased to inform you that your manuscript has been deemed suitable for publication in PLOS ONE. Congratulations! Your manuscript is now with our production department. 

Kind regards, 

on behalf of

Dr. Faisal Abbas 

Academic Editor

PLOS ONE